# Entropy: An Inspiring Tool for Characterizing Turbulence–Combustion Interaction in Swirling Flames via Direct Numerical Simulations of Non-Premixed and Premixed Flames

**DOI:** 10.3390/e25081151

**Published:** 2023-08-01

**Authors:** Jingke Su, Anxiong Liu, Hualin Xiao, Kun Luo, Jianren Fan

**Affiliations:** 1State Key Laboratory of Clean Energy Utilization, Zhejiang University, Hangzhou 310027, China; sujingke@zju.edu.cn (J.S.); hulonxiao@zju.edu.cn (H.X.); zjulk@zju.edu.cn (K.L.); fanjr@zju.edu.cn (J.F.); 2Shanghai Institute for Advanced Study, Zhejiang University, Shanghai 200000, China

**Keywords:** direct numerical simulation, entropy generation, turbulence–combustion interaction

## Abstract

This article focuses on entropy generation in the combustion field, which serves as a useful indicator to quantify the interaction between turbulence and combustion. The study is performed on the direct numerical simulations (DNS) of high pressure non-premixed and premixed swirling flames. By analyzing the entropy generation in thermal transport, mass transport, and chemical reactions, it is found that the thermal transport, driven by the temperature gradient, plays a dominant role. The enstrophy transport analysis reveals that the responses of individual terms to combustion can be measured by the entropy: the vortex stretching and the dissipation terms increase monotonically with the increasing entropy. In high entropy regions, the turbulence behaves as the “cigar shaped” state in the non-premixed flame, while as the axisymmetric state in the premixed flame. A substantial increase in the normal Reynolds stress with the entropy is observed. This is due to the competition between two terms promoted by the entropy, i.e., the velocity–pressure gradient correlation term and the shear production term. As a result, the velocity–pressure gradient correlation tends to isotropize turbulence by transferring energy increasingly from the largest streamwise component to the other smaller normal components of Reynolds stress and is dominated by the fluctuating pressure gradient that increases along the entropy. The shear production term increases with the entropy due to the upgrading alignment of the eigenvectors of strain rate and Reynolds stress tensors.

## 1. Introduction

Swirling flames are widely used in a variety of combustion applications, including gas turbines, aero-engines, and internal combustion engines, which always operate in high pressure and temperature conditions [1,2]. The strong non-linear interaction between turbulence and combustion is one of the remarkable features of swirling flames. Swirling structures have been shown to intensify the heat release [3], short the ignition delay [4], influence the extinction [5], and control the emissions [6] of the flames. Good understanding of the turbulent dynamic interaction with intense combustion is desirable to enable improved design of efficiency.

Much previous work on the non-linear and multi-scale interaction between turbulence and combustion has been done for the improvement of turbulent combustion models and combustion efficiency, including on the effect of turbulence on flame structures [7], flame speed [8], and flame thickness. Innovatively, flame-generated turbulence was first indicated by Karlovitz et al. [9], and, subsequently, Bray et al. [10] found that flame induces thermal expansion and accelerates the flow velocity in the flame-normal direction, which is experimentally confirmed by Moreau and Boutier [11]. Recently, an increasing amount of research has been performed to understand the two-way coupling interaction between turbulence and combustion in swirling flames. Earlier research illustrated that flow structures in a typical swirling flame usually contained an inner recirculation zone, an annular swirling jet, and an outer recirculation zone, and their performances were significantly influenced by intense flame [2,12]. Sirui Wang et al. [13] revealed that the flame-induced flow acceleration near the burner outlet enhanced the vorticity and velocity in a stratified swirling flame, while the length of the recirculation zone was found to be smaller than that without reaction. The increasing back-scatter turbulent kinetic energy transfer was observed internal to the swirling flames instead of down-scale transfer in the non-reacting regions [14]. The experiment measured that the enstrophy production was determined by combustion and turbulence properties and the flame-induced baroclinic turbulence production should be considered [15].

Direct numerical simulation (DNS) is a promising approach for completely resolving all of the relevant turbulence and chemical scales both in space and time. It provides access to full three-dimensional, time-varying scalars and velocity that are being transported, and represents a useful tool to provide insights for interaction between turbulence and combustion. Xiao et al. [16,17,18] performed DNS of non-premixed and premixed swirling flames and found that combustion accelerated radial velocity and suppressed the production of turbulent kinetic energy (TKE) in the shear layers. Moreover, the turbulence in the map of Reynolds-stress anisotropy left the axisymmetric state and moved towards the “cigar shaped” state due to the heat release. DNS of hydrogen/air swirling premixed flame demonstrated that the flame front was entrained by large-scale coherent structures, resulting in larger coherent structures near the inlet and small-scale vortical structures further upstream [19]. Furthermore, an important study is the decoupling of interaction between turbulence and combustion in the corrugated flamelets and thin reaction zones regimes via the Karlovitz number [20]. The local flame thickness in the corrugated flamelets regime remains smaller than the Kolmogorov length scale so that the turbulent eddies do not affect the flame structures and vice versa in the thin reaction zones regime [20]. TKE within the corrugated flamelets regime is generated by the flame brush. However, it has been found that the TKE decays monotonically across the flame brush within the thin reaction zones regime [21]. Similarly, MacArt et al. [22] defined a critical Karlovitz number to elucidate the effect of heat release on the turbulence, showing that the flame-induced velocity–pressure gradient increased and dominated TKE production and thermal expansion misaligned the invariants of the anisotropic Reynolds stress and the strain rate tensor with each other. Subsequently, Lee et al. [23] analyzed the effect of heat release on the Reynolds stresses transport. A significant increase was found in the flame-normal component of the normal Reynolds stresses, and the velocity–pressure gradient in the Reynolds stress budgets served fundamentally different roles at low and high Karlovitz numbers.

The above-mentioned studies have suggested two methods for understanding the interaction between turbulence and combustion. The first method is the direct comparison between reacting and non-reacting flows by turning off the chemical reaction. However, its shortcoming is that it does not take into account the two-way coupling interaction between turbulence and combustion. The second type is to decouple turbulence or combustion from the two-way interaction. For example, reactor-based models and flame front models utilize a quasi-laminar flame approximation and exclude the influence of turbulence eddies on the smallest-scale flame structure in order to predict the chemical reaction rates within the corrugated flamelets regime. On the other hand, in thin reaction zones with a high Karlovitz number, the influence of combustion on the turbulent flow vortices is neglected (the Boussinesg model [22,23]). Nevertheless, a practical combustion process often encompasses multiple regimes, making them impossible to be covered by the decoupling models. Therefore, in this study, we introduce the measures of entropy to identify the two-way turbulence–combustion interaction. In addition, the terms “reaction progress variable” and “mixture fraction” are commonly used in combustion systems to characterize the location of maximum heat release and flame front. However, neither of these variables quantitatively describe the impact of combustion on flow fields. The location of the flame front with the highest heat release does not necessarily correspond to the region where combustion has the greatest impact on the flow field [24,25]. The disruption of the equilibrium state in a combustion system is primarily caused by thermodynamic potentials quantified by the entropy rather than heat release.

Now let us in a general thermodynamic issue view the combustion process from a different perspective: a number of coupled physical processes that trigger and support the combustion process in the combustion system can be classified into two groups, i.e., transport processes and chemical reactions, both of which are driven by the thermodynamic potentials for the fluxes of mass, momentum, and energy. The potentials are defined by the gradients of different variables, such as temperature, velocity, chemical affinity, and species concentration in a combustion system, all of which are characterized by thermodynamic irreversibility [24]. The evolution of a combustion system is inherently driven by these irreversible potentials, resulting in physical processes that can be measured using thermodynamic entropy. Therefore, entropy can be utilized as a tool to characterize these physical processes and assess their role in the evolution of the combustion system. This characterization can provide insights into how the thermodynamic potentials caused by combustion influence the flow fields. The local entropy generation is a crucial physical quantity that reveals the level of local thermodynamic irreversibility in a combustion system. It serves as an important criterion for indicating how to enhance useful irreversible processes and prevent harmful irreversible processes. For example, the performance of a compressor stage design increases with the growth of total pressure (representing useful irreversibility), but its efficiency decreases due to pressure loss (representing harmful irreversibility). Quite a lot of research has focused on entropy generation to improve the useful part of entropy generation by exploring the effects of boundary condition type [26], soot, flame temperature [27], and fuel blends [28] on local entropy generation rate.

In this work, we perform DNS of the non-premixed and premixed swirling flames, and the objective is to improve our understanding of the mechanism of interaction between turbulence and combustion in terms of thermodynamic entropy. To the best of the authors’ knowledge, this is first time that the entropy is used to quantitatively characterize the two-way interaction between turbulence and combustion. The paper is organized as follows. Section 2 describes the DNS configuration and mathematical methods employed in the present study, including the entropy transport equation. The DNS results are presented and discussed in Section 3, which focuses on the interaction of vorticity and Reynolds stress with combustion investigated by the entropy. Finally, conclusions are made in Section 4.

## 2. Mathematical Methods and DNS Configuration

### 2.1. Governing Equations and Numerical Methods

The DNS is performed by a low Mach number in-house code [29], which solves the compressible conservation equations of mass, momentum, energy, and chemical species by finite difference methods. In addition, the equation of state for ideal gas is included,
(1)$$\frac{\partial \rho }{\partial t}+\frac{\partial \rho u_{i}}{\partial x_{i}}=0$$
(2)$$\frac{\partial \rho u_{i}}{\partial t}+\frac{\partial \rho u_{i}u_{j}}{\partial x_{j}}=-\frac{\partial p}{\partial x_{i}}+\frac{\partial \tau_{ij}}{\partial x_{j}}$$
(3)$$\begin{array}{c} c_{p}\left(\frac{\partial \rho T}{\partial t}+\frac{\partial \rho u_{j}T}{\partial x_{j}}\right)=\nabla \cdot \left(\rho c_{p}\lambda \nabla T\right)+\\ \sum_{k}\rho c_{p,k}\left(D_{k}\nabla Y_{k}+Y_{k}{\mathrm{Vc}}^{k}\right)\cdot \nabla T+{\dot{\omega }}_{T} \end{array}$$
(4)$$\frac{\partial \rho Y_{k}}{\partial t}+\frac{\partial \rho u_{j}Y_{k}}{\partial x_{j}}=\nabla \cdot \left(\rho D_{k}\nabla Y_{k}\right)+\nabla \cdot \left(\rho Y_{k}V_{k}\right)+{\dot{\omega }}_{k}$$
where ρ is the density, *u* is the velocity, $\tau_{ij}$ is the viscous stress tensor, $c_{p}$ is the mixture heat capacity, $c_{p,k}$ is the species heat capacity, *T* is the temperature, λ is the mixture-averaged specific thermal conductivity, $D_{k}$ is the mass diffusivity, $Y_{k}$ is the species mass fraction, and $\dot{w_{k}}$ and $\dot{w_{T}}$ are the source terms of species and temperature, respectively. The $V_{k}$ represents the corrected diffusivity of species *k* to ensure mass conservation.

The code applies a third-order non-oscillatory (WENO) scheme to the scalar-transport equations. Time advancement is achieved through a semi-implicit Crank–Nicolson method. At the walls, the non-slip adiabatic boundary condition is utilized, and heat flux is ignored between the pilot and main stages. The inflow boundary condition is based on the simulation of a fully developed turbulent pipe flow. The outlet employs a non-reflective boundary condition. A published reduced chemical mechanism for kerosene/air is employed, which contains two-step reactions and six species [30]. This mechanism has been validated against the detailed mechanism to predict the laminar flame speed. It also accurately predicts the ignition delay time for initial temperatures ranging from 900 K to 1500 K, operating at stoichiometric mixture and a pressure of 20 bar.

### 2.2. The Transport Equation of Entropy

The general specific entropy *s* transport equation for compressible Newtonian fluids is given, which is obtained from the mass and energy conservation, following Hirschfelder et al. (1954) [31], as,
(5)$$T\rho \frac{Ds}{Dt}=\rho \frac{De}{Dt}+p\rho \frac{Dv}{Dt}-\sum_{k=1}^{N}\mu_{k}\rho \frac{DY_{k}}{Dt}$$
where *e*, *v*, *p*, and $\mu_{k}$ are the internal energy, specific volume, pressure, and specific chemical potential of species *k*. For an ideal gas, the entropy transport equation can be written from conservation equations of energy and chemical species,
(6)$$\begin{array}{c} \frac{\partial \rho s}{\partial t}+\frac{\rho u_{j}s}{x_{j}}=\frac{1}{T}\left[\tau_{ij}\frac{\partial u_{i}}{\partial x_{j}}-\frac{1}{T}\left(q_{j}-\sum_{k=1}^{n}h_{k}q_{k,j}\right)\frac{\partial T}{\partial x_{j}}+\right.\\ \left.\sum_{k=1}^{n}q_{k,j}\left(\frac{\partial \mu_{k}}{\partial x_{j}}+s_{k}^{0}\frac{\partial T}{\partial x_{j}}\right)-\sum_{k=1}^{n}\mu_{k}\dot{w_{k}}\right]-\\ \frac{\partial }{\partial x_{j}}\left[\frac{1}{T}\left(q_{j}-\sum_{k=1}^{n}h_{k}q_{k,j}\right)+\sum_{k=1}^{n}s_{k}^{0}q_{k,j}\right] \end{array}$$
where the heat flux $q_{j}$ and diffusive flux $q_{k,j}$ can be expressed using Fourier’s law of heat conduction and Fick’s law of mass diffusion,
(7)$$q_{j}=-\lambda \frac{\partial T}{\partial x_{j}}+\sum_{k=1}^{n}h_{k}q_{k,j}$$
(8)$$q_{k,j}=-\rho D_{k}\frac{\partial Y_{k}}{\partial x_{j}}$$
here, the Soret (molecular species diffusion due to temperature gradients) and Dufor (heat flux due to species mass fraction gradients) effects in hydrocarbon-air flames can be neglected [32,33], which have been ignored in many pieces of research including other DNS works [34,35,36]. The terms of the RHS of Equation (Equation 6) can be expressed as,
(9)$$\sum_{k=1}^{n}q_{k,j}\left(\frac{\partial \mu_{k}}{\partial x_{j}}+s_{k}^{0}\frac{\partial T}{\partial x_{j}}\right)=\sum_{k=1}^{n}\left(q_{k,j}\frac{c_{p,k}}{T}\frac{\partial T}{\partial x_{j}}-q_{k,j}\frac{\partial s_{k}^{0}}{\partial x_{j}}\right)$$
(10)$$\frac{\partial s_{k}^{0}}{\partial x_{j}}=\frac{c_{p,k}}{T}\frac{\partial T}{\partial x_{j}}-\frac{R_{k}}{X_{k}}\frac{\partial X_{k}}{\partial x_{j}}-\underset{\Theta }{\underset{︸}{\frac{R_{k}}{p}\frac{\partial p}{\partial x_{j}}}}$$
(11)$$\begin{array}{c} \frac{\partial }{\partial x_{j}}\left[\frac{1}{T}\left(q_{j}-\sum_{k=1}^{n}h_{k}q_{k,j}\right)+\sum_{k=1}^{n}s_{k}^{0}q_{k,j}\right]=-\frac{\partial }{\partial x_{j}}\left(\frac{\lambda }{T}\frac{\partial T}{\partial x_{j}}+\sum_{k=1}^{n}\rho s_{k}^{0}D_{k}\frac{\partial Y_{k}}{\partial x_{j}}\right) \end{array}$$
where $X_{k}$, $M_{k}$, $R_{k}$, and $s_{k}^{0}$ are the mole fraction, molecular weight, gas constant, and standard-state specific entropy of the pure species *k*. The last term Θ of Equation (Equation 10) can be neglected for low Mach number flows [25,37,38], and the specific heat capacities and mass diffusivities of different species are taken to be identical, as well as unity Lewis number [36,39],
(12)$$\begin{array}{c} \frac{\partial \rho s}{\partial t}+\frac{\partial \rho u_{j}s}{\partial x_{j}}=\nabla \cdot \left(\rho D\nabla s\right)+{\dot{S}}_{\mathrm{vis}}+{\dot{S}}_{\mathrm{heat}}+{\dot{S}}_{\mathrm{mass}}+{\dot{S}}_{\mathrm{reac}} \end{array}$$
(13)$${\dot{S}}_{\mathrm{vis}}=\frac{\tau_{ij}}{T}\frac{\partial u_{i}}{\partial x_{j}}$$
(14)$${\dot{S}}_{\mathrm{heat}}=\frac{\lambda }{T^{2}}\nabla T\cdot \nabla T$$
(15)$${\dot{S}}_{\mathrm{mass}}=\sum_{k=1}^{N}\rho D_{k}R_{k}\nabla X_{k}\cdot \nabla Y_{k}$$
(16)$${\dot{S}}_{\mathrm{reac}}=-\frac{1}{T}\sum_{k=1}^{N}\frac{\mu_{k}\dot{w_{k}}}{M_{k}}$$
the last four terms represent entropy generation due to viscous dissipation ${\dot{S}}_{\mathrm{vis}}$, thermal transport ${\dot{S}}_{\mathrm{heat}}$, mass transport ${\dot{S}}_{\mathrm{mass}}$, and chemical reaction ${\dot{S}}_{\mathrm{reac}}$. The sum of these four terms is the total entropy generation rate ${\dot{S}}_{\mathrm{gen}}$. A similar form of the entropy transport equation (Equation (Equation 12)) was analyzed in the DNS context [26,36,39,40], which was emphasized as a useful tool to develop and improve LES/RANS models [38,41,42].

In this work, Equation (Equation 12) is not mandatory to be solved, as entropy $S_{k}$ of the pure species *k* is a fitting function of the temperature, pressure, and species concentration,
(17)$$S_{k}=S_{k}^{0}-R\ln X_{k}-R\ln\left(P/P_{\mathrm{atm}}\right)$$
(18)$$S_{k}^{0}/R=a_{1k}\ln T_{k}+a_{2k}T+\frac{a_{3k}}{2}T_{k}^{2}+\frac{a_{4k}}{3}T_{k}^{3}+\frac{a_{5k}}{2}T_{k}^{4}+a_{7k}$$
where $S_{k}^{0}$ is the standard-state entropy of the pure species *k*. The absolute value of entropy can be calculated from the DNS framework by solving Equations (Equation 1)–(Equation 4). The local entropy inside the combustion system is changing due to the processes of convection, diffusion, and flux through the fluid boundary and the entropy generation [41]. The convective and diffusion terms of Equation (Equation 12) are responsible for redistributing the entropy flux and do not increase the total entropy in the system. As in the low-Mach flows, the unsteady entropy wave through the inlet and the outlet can be neglected so that the contributions of the total entropy to the system are balanced by the steady inlet fulfilling the air and fuel and the convective outlet. Together with the adiabatic walls, these boundary conditions do not increase the total entropy of the open system. In this work, we only focus on analyzing the source terms of the entropy generation characterizing thermodynamic irreversibility. The entropy generation can be fully calculated from the DNS results, which are dependent on the statistics of reaction rate, $\dot{w_{k}}$, and scalar gradients, $\nabla T,\nabla Y_{k},\nabla X_{k}$, according to Equation (Equation 13)–(Equation 16).

### 2.3. DNS Configuration

The DNS of high pressure non-premixed (NP)/premixed (PR) swirling flames based on the simplified Tecflam swirl burner [43,44] are performed. Figure 1 shows the cross sections of the computational domains, which were described in detail in our previous work [17]. For the non-premixed flame, the inner diameters of the fuel and air inlet are 0.06 mm and 0.09 mm ($R_{i}$), and the outer diameters $\left(R_{0}\right)$ are 0.08 mm and 0.18 mm, respectively. The inner annulus serves as the fuel inlet, with a velocity of 1.17 m/s and a temperature of 300 K. The outer annulus acts as the air inlet, with a velocity of 40 m/s and a temperature of 760 K. The global equivalence ratio of 0.6 operates at a pressure of 20 bars. In the premixed case, the inner diameter of the annular inlet is 0.05 mm, the outer diameter is 0.15 mm, the global equivalence ratio is 0.7, inlet velocity is set to 40 m/s, and temperature is set to 760 K. The inlet bulk Reynolds number is set to 4000, and the swirling number of air flow is 1.0. These cases serve to mimic aero-engine conditions.

The burners are the cylindrical structures, the length of the computational domains in the streamwise is $5D\;\left(D=2R_{0}\right)$, and the radial width is 3D. A uniform mesh is used in the streamwise, *x*, and circumferential directions, θ, while a stretched mesh with algebraic stretching rate of 1.01 is used in the radial direction, *r*. The grids near the inlet, the center line, the shear layer, and the wall are locally refined to ensure the simulation quality. The minimum size of grid cells is 0.7 μm. The Kolmogorov length scale, which is defined as $\eta ={\left(\frac{{\tilde{\nu }}^{3}}{\tilde{\epsilon }}\right)}^{\frac{1}{4}}$, has a minimum value of 1.12 μm near the fuel nozzle, where ν is the kinetic viscosity and ϵ is the turbulent dissipation rate. More than ten grid cells across the flame thermal thickness $\delta_{L}$ [45] and the criterion Δ/η>2 [46] is satisfied to fully resolve both turbulence and flame structures. The time step size ranges from 3 ns to 12 ns to satisfy the condition of the Courant–Friedrichs–Levy (CFL) number less than 0.3 [47]. The number of grid points in the non-premixed case is $N_{x}\times N_{r}\times N_{\theta }=668\times 256\times 256$, and that in the premixed case is $N_{x}\times N_{r}\times N_{\theta }=668\times 384\times 256$.

## 3. Results and Discussion

The local distributions of entropy generation in the two swirling flames are investigated in Section 3.1, and the individual terms can be used to quantitatively describe the physical process in the combustion systems. In Section 3.2, the entropy generation due to thermal transport provides a measure of the intensity of thermal expansion, and the interaction of combustion with enstrophy transport and turbulence anisotropy is therefore quantified. In Section 3.3, the Reynolds stress and its budget conditionally averaged on the entropy are analyzed, and then the responses of primary source and sink to combustion are discussed through the analysis of entropy.

### 3.1. Statistical Behaviors of Entropy Generation

The dynamics and stabilization of a flame are significantly influenced by the flow structures in a swirling flame [13]. As the flame behavior plays a crucial role in shaping entropy generation, it is necessary to investigate the flow fields, with the aim of gaining a comprehensive understanding of their impact on flame and entropy generation. The Favre averaged axial velocity (${\tilde{u}}_{x}$) on the center plane for the non-premixed (NP) and premixed (PR) flames is shown in Figure 2, which is used to characterize the five typical swirling flow structures [16,17]. The recirculation zone is defined as a structure where the mean axial velocity is negative. The inner recirculation zone $\left(\mathrm{IRZ}\right)$ is formed by vortex breakdown in the central region. The outer recirculation zone $\left(\mathrm{ORZ}\right)$ is formed by the strong swirling effects and confinement, leading to continuous reverse flow [48]. These two zones carrying hot burnt gas from downstream to upstream significantly influence flame stabilization. The region with the maximum mean axial velocity $\left[{\tilde{u}}_{x}\left(x,r\right)\geq 0.5{\tilde{u}}_{x}{\left(r|x\right)}_{\max}\right]$ is the main swirling zone $\left(\mathrm{MSZ}\right)$, with the inner shear layers $\left(\mathrm{ISL}\right)$ on the inner side and the outer shear layers $\left(\mathrm{OSL}\right)$ on the outer side. The large-scale vortices generated in shear layers directly interact with flames, primarily contributing to the internal instability [16,17].

The distributions of the local entropy generation are dominated by the flow structures and heat release. Figure 3a,b shows the Reynolds-averaged entropy generation and heat release on the center plane, x−r, where the black velocity streamlines denote the flow trajectory. In both cases, the value of entropy generation spans over several orders of magnitude in high heat release regions, which are highlighted in a log-scale. It can be observed that the discrepancies in the heat release result in the different distribution of entropy generation. The heat release in the case of NP is much higher than that in the case of PR, leading to the presence of higher entropy generation. The local entropy generation in the upstream is large and decays along the streamlines in magnitude, but the distribution increasingly broadens in the downstream. The streamlines indicate that the entropy generation is strong in shear layers, while weak in recirculation zones.

Figure 3a shows that the entropy generation of the NP case mainly occurs in the ISL, which peaks in the zones extending around the nozzle outlet and decreases rapidly along the radial direction. The narrow distributions widen slightly as there is increasing mixing of fuel and oxidant along the streamwise direction and then decays rapidly, mainly due to the significant consumption of fuel at downstream locations. The recirculation zone due to the strong swirling motion continuously transports the hot burnt gas from downstream to upstream, resulting in a decrease of entropy generation by at least one order of magnitude along the streamwise direction. Therefore, the distributions of the heat release and entropy generation are not fully overlapped. Additionally, it demonstrates the presence of significant entropy generation inside the OSL. In the case of PR, the distribution of heat release shows better agreement with the entropy generation compared to the NP case. Figure 3b shows that high levels of heat release and entropy generation are exhibited in the ISL and OSL, but the entropy generation in the OSL is higher than in the ISL. The entropy generation starts expanding rapidly from the inlet due to the strong swirling shear force, and the distributions of entropy generation are wider than those in the case of NP. Note that the entropy generation in the case of PR decreases relatively slowly along the streamwise direction, and the volume average of the total entropy generation ($1.5\times 10^{8}$J/(kmol·K)) is lower than in the case of NP ($2.2\times 10^{8}$J/(kmol·K)).

The mixture fraction $Z_{\mathrm{mix}}$ is defined at any location in the system as the local ratio of the mass flux originating from the fuel fed to the sum of both mass fluxes. $Z_{\mathrm{mix}}$ can be derived from the C, H, and O elements as follows [49],
(19)$$Z_{\mathrm{mix}}=\frac{2Y_{C}/W_{C}+\frac{1}{2}Y_{H}/W_{H}+\left(Y_{O,2}-Y_{O,2}\right)/W_{O}}{2Y_{C,1}/W_{C}+\frac{1}{2}Y_{H,1}/W_{H}+Y_{O,2}-Y_{O,2}/W_{O}}$$
where $Y_{j}$ and $W_{j}$ are the elemental mass fractions and atomic masses for the elements carbon, hydrogen, and oxygen, and the subscripts 1 and 2 refer to values in the fuel and air streams, respectively. Figure 4a shows the entropy generation budget and heat release conditionally averaged on ${\tilde{Z}}_{mix}$ for the case of NP. It is obvious that the entropy generation profiles in the non-premixed flame have two peaks because there are two combustion modes, i.e., the non-premixed combustion mode with the mixture fraction $0.03<{\tilde{Z}}_{\mathrm{mix}}<0.045$ and the premixed combustion mode with the mixture fraction ${\tilde{Z}}_{\mathrm{mix}}>0.07$ [50]. It can be observed that the premixed combustion mode is the main contributor to the entropy generation, where oxygen is transported to fuel on the fuel-rich side.

The reaction progress variable C is defined based on the mass fraction of $O_{2}$, which can be derived as follows [21],
(20)$$C=\left(Y_{O_{2}}-Y_{O_{2},u}\right)/\left(Y_{O_{2},b}-Y_{O_{2},u}\right)$$
where $Y_{O_{2},u}$ is the $O_{2}$ mass fraction of the reactants, while $Y_{O_{2},b}$ is the $O_{2}$ mass fraction of fully burned products. According to this definition, the progress variable is a step function that separates unburnt mixture and burnt gas in a given flow field, which rises monotonically from zero in fresh reactants to unity in fully burned products. In the premixed case, the entropy generation budget and heat release evaluated at the reaction progress variable are shown in Figure 4b. It is obvious that the entropy generation is mainly distributed at $0.3<\tilde{C}<0.4$, but the chemical reaction process mainly occurs in the fully burned regions near $\tilde{C}=0.8$. This evidence illustrates that the thermodynamic process of thermal transport, mass transport, and chemical reaction are not fully coupled, which is consistent with Stanciu et al. [25].

The budget of various terms on the RHS of Equation (Equation 12) is shown in Figure 5a,b at upstream $x=R_{0}$. The shadows in Figure 5 represent the typical regions, which, from left to right, are the IRZ, ISL, MSZ, OSL, and ORZ. The $x=R_{0}$ corresponds to the positions where the entropy distribution is well developed and not considerably decreased or confined by the wall from Figure 3. Additionally, the overall trend observed in the budget is related to the flow structures, which is conducive to understanding the evolution of the flow field. In both cases, the thermal transport $\left({\tilde{\dot{S}}}_{\mathrm{heat}}\right)$ is the primary source in the entropy generation budget, while the value of the viscous dissipation $\left({\tilde{\dot{S}}}_{\mathrm{vis}}\right)$ can be ignored and will not be discussed.

For the case of NP, the thermal transport term $\left({\tilde{\dot{S}}}_{\mathrm{heat}}\right)$ and the total entropy generation reach the maximum in the ISL, where the flame fully consumes reactants, resulting in the high intensity heat release. The thermal transport in this region is stronger than that in other regions due to the high temperature gradient field caused by the high intensity heat release. However, with the rapid consumption of reactants, the species concentration gradients decrease rapidly, leading to the reduction of the transport rate of reactants to the flame indicated by the mass transport term $\left({\tilde{\dot{S}}}_{\mathrm{mass}}\right)$. Therefore, the heat release intensity also decreases, and thus the terms of the entropy generation budget decrease significantly outside the ISZ. Unlike in the case of NP, the entropy generation in the ISL in the case of PR is relatively low. These terms mainly generate in the IRZ and MSZ because the flame areas are more widely distributed in the flow structures.

To further clarify the relationship between the entropy generation and the flow, the largest source $\left({\tilde{\dot{S}}}_{\mathrm{heat}}\right)$ in Equation (Equation 12) after Favre average can be decomposed into the resolved component,
(21)$${\tilde{\dot{S}}}_{\mathrm{heat}}^{\mathrm{resol}}=\left(\tilde{\lambda }\nabla \tilde{T}\cdot \nabla \tilde{T}\right)/{\tilde{T}}^{2}$$
and the unresolved components aside from ${\tilde{\dot{S}}}_{\mathrm{heat}}^{\mathrm{resol}}$,
(22)$${\tilde{\dot{S}}}_{\mathrm{heat}}^{\mathrm{unresol}}={\tilde{\dot{S}}}_{\mathrm{heat}}-{\tilde{\dot{S}}}_{\mathrm{heat}}^{\mathrm{resol}}$$In both cases, the contribution of turbulent fluctuations (see Figure 6a,b) is the main contribution of entropy generation, and even the influence of the average field can be ignored. In addition, the entropy generation $\left({\tilde{\dot{S}}}_{\mathrm{heat}}\right)$ is primarily directly affected by the temperature gradient factor ∇T·∇T and via an indirect effect of $\lambda /T^{2}$. Then, the entropy generation components in the coordinate direction, i.e., ${\tilde{\dot{S}}}_{\mathrm{heat},x}$, ${\tilde{\dot{S}}}_{\mathrm{heat},r}$, and ${\tilde{\dot{S}}}_{\mathrm{heat},\theta }$, are also shown in Figure 6a,b. The mean entropy generation in the circumferential direction ${\tilde{\dot{S}}}_{\mathrm{heat},\theta }$ is lower than that in the other two directions. In the case of NP, the mean entropy generation in the streamwise direction ${\tilde{\dot{S}}}_{\mathrm{heat},x}$ is close to that in the radial direction ${\tilde{\dot{S}}}_{\mathrm{heat},r}$ because the thermal transport intensity is high in both directions. The difference in the case of PR is that the entropy generation ${\tilde{\dot{S}}}_{\mathrm{heat}}$ depends primarily on the radial direction ${\tilde{\dot{S}}}_{\mathrm{heat},r}$, where the thermal transport in this direction is relatively strong. This observation indicates that the flame area distribution in the case of PR is broader than that in the case of NP.

The above analysis shows that the main source of irreversibilities in both types of swirling burners is the thermal transport process associated with high temperature gradients caused by heat release. Therefore, the main method to optimize the distribution of entropy generation is to properly control the temperature field and thermal transport process, creating a flow field that optimizes the distribution of entropy generation. Some scholars have achieved a better distribution of entropy generation, for example, by adding active components [28] to the reaction flow, optimizing boundary conditions [26], adopting a multi-objective method [51], etc.

Figure 3, Figure 4, Figure 5 and Figure 6 illustrate that the two cases result in the distinct distributions of the entropy generation. This consequence can be better understood by the entropy generation due to the chemical reaction process $\left({\tilde{\dot{S}}}_{\mathrm{reac}}\right)$ determined by the reaction rate and the chemical affinity (the ability to drive the chemical reaction). The contribution of different species appearing in the chemical reaction to the evolution of ${\tilde{\dot{S}}}_{\mathrm{reac}}$ is illustrated in Figure 7a,b. The magnitude of entropy generation is negative in the reactants while is positive in the products.

In the case of NP, the ${\tilde{\dot{S}}}_{\mathrm{reac}}$ and its components are most significant in the ISL and significantly decrease in other regions due to the depletion of reactants. The flame area is therefore narrowly distributed in the ISL, where kerosene significantly consumes oxygen and produces a large oxygen concentration gradient. This is responsible for transporting oxygen from the MSZ and IRZ to the ISL, where the reactants are almost completely consumed, resulting in the narrow flame distributions. However, the reactions mainly occur in the OSL in the case of PR. The hot products with high temperature are entrained into the IRZ region at farther downstream positions, which produce higher temperature gradient resulting in wider distribution and a great magnitude of entropy generation. From Figure 5, the chemical reaction process and the transport processes do not always take place in the same regions.

### 3.2. Interaction between Vorticity Dynamics and Combustion Investigated by Entropy

From the last section, we can deduce that the reaction progress variable $\tilde{C}$ and mixture fraction ${\tilde{Z}}_{\mathrm{mix}}$ representing the progress of the combustion system are not equivalent to the “force” driving the system from the thermodynamic view. The generalized thermodynamics “force” represented by temperature gradient, chemical affinity, and species gradients leads to the generalized thermodynamics “flux” represented by thermal transport, chemical reaction, and mass transport, which drives the evolution of combustion systems. Furthermore, the chemical reaction process and the transport processes do not always occur in the same regions, especially in swirling flames with the dominant flow structures.

The evolution of combustion systems must be accompanied by the mentioned “force” and “flux”, and the relevant physical processes can be measured much better by the entropy generation analysis. In both cases, the dominant term for thermal transport $\left({\tilde{\dot{S}}}_{\mathrm{heat}}\right)$ in the entropy generation budget illustrates the significant interaction of the flame-induced temperature gradient (“force”) and the subsequent thermal transport process (“flux”) with turbulent structures, thereby characterizing the impact of combustion on the flow field. Therefore, the entropy generation due to the thermal transport is used to quantitatively analyze the interaction between turbulence and combustion.

The effect of combustion on mean and fluctuating velocity was discussed in the literature [16,17,18], and it was found that combustion induced flow dilatation and accelerated mean velocity. In the shear layers, the TKE of non-premixed flame was suppressed instead of a significant increase in premixed flame. In this section, the interaction of vorticity dynamics with combustion will be explained more.

The enstrophy $\omega^{2}=\omega_{i}\omega_{i}$ is an important characteristic in vorticity dynamics, and its transport equation is as follows [17,52],
(23)$$\frac{1}{2}\frac{D\omega^{2}}{Dt}=\underset{V_{1}}{\underset{︸}{\omega \cdot S\cdot \omega }}\underset{V_{2}}{\underset{︸}{-\omega^{2}\left(\nabla \cdot u\right)}}+\underset{V_{3}}{\underset{︸}{\frac{\omega }{\rho^{2}}\cdot \left(\nabla \rho \times \nabla p\right)}}+\underset{V_{4}}{\underset{︸}{\omega \cdot \nabla \times \left(\frac{1}{\rho }\nabla \cdot \tau \right)}}$$
on the RHS of the equation, there are vortex stretching $\left(V_{1}\right)$, dilatation $\left(V_{2}\right)$, baroclinic torque $\left(V_{3}\right)$, and correlation terms of viscous transport and viscous dissipation $\left(V_{4}\right)$ terms. The subsequent analysis of turbulence statistics focuses on the location $x=1.5R_{0}$, signified as the location of the maximum heat release [16,17,18]. Figure 8 shows the budget of various terms in Equation (Equation 23) conditionally averaged on ${\tilde{\dot{S}}}_{\mathrm{heat}}$. In both cases, Equation (Equation 23) shows that vortex stretching $\left(V_{1}\right)$ is the primary source of enstrophy, which increases monotonically with ${\tilde{\dot{S}}}_{\mathrm{heat}}$. This demonstrates that combustion promotes the increase of vortex stretching $\left(V_{1}\right)$, thereby increasing enstrophy. The dissipation term $\left(V_{4}\right)$ in Equation (Equation 23) acts as the predominant sink, exhibiting a substantial magnitude that increases with ${\tilde{\dot{S}}}_{\mathrm{heat}}$. This demonstrates that the effects of viscous transport and viscous dissipation increase with combustion effects, playing a dominant role in dissipating and reducing enstrophy. The balance between these two terms determines the balance between production and dissipation of enstrophy. It is also seen that the responses of the dilatation term $\left(V_{2}\right)$ and the baroclinic torque term $\left(V_{3}\right)$ to ${\tilde{\dot{S}}}_{\mathrm{heat}}$ increase with entropy, but their magnitudes are small. The dilatation $\left(V_{2}\right)$ shows that combustion enlarges the radius of the small-scale vortex, leading to a decrease in enstrophy. On the other hand, the increasing baroclinic torque $\left(V_{3}\right)$ contributes to the generation of enstrophy. Note that the mean profiles of the vortex stretching $\left(V_{1}\right)$ and dissipation $\left(V_{4}\right)$ terms in the case of PR increase more significantly in high ${\tilde{\dot{S}}}_{\mathrm{heat}}$ regions than in low ${\tilde{\dot{S}}}_{\mathrm{heat}}$ regions. This is because the wider distribution of the PR flame leads to sufficient interaction with different fluid structures.

The above analysis indicates that the various terms in the enstrophy transport and the entropy are strongly correlated. Therefore, Figure 9a,b shows the joint PDF of the vortex stretching $\left(V_{1}\right)$ and ${\tilde{\dot{S}}}_{\mathrm{heat}}$ (Figure 9a), the dissipation $\left(V_{4}\right)$ and ${\tilde{\dot{S}}}_{\mathrm{heat}}$ (Figure 9b), and the conditional means of each quantity on ${\tilde{\dot{S}}}_{\mathrm{heat}}$ at $x=1.5R_{0}$. Figure 9a shows that high positive vortex stretching regions generally have great entropy generation, and low vortex stretching regions occur most often in low entropy generation regions. However, the negative vortex stretching is observed in low entropy generation regions, which are balanced by their higher positive values, resulting in positive conditional averages on the entropy. The response of the vortex stretching to entropy depends on the effect of flames upon the alignment relationship between the vorticity vector and the eigenvectors of strain rate tensor [17]. Figure 9b shows that the net negative strong dissipation regions consistently occur in the regions that have high entropy generation and decrease monotonically along the entropy generation.

Not only is the vorticity field influenced by the local interaction of thermal transport, but both large- and small-scale fluid structures are also altered by the flames. The behavior of anisotropic fluid structures can be described by the Reynolds stress invariants. Since the trace of the normalized Reynolds stress anisotropy tensor $b_{ij}$ is zero, there are only two invariants [17,53]: the second invariant II and the third invariant III, which are given by
(24)$$II=-\frac{b_{ij}b_{ji}}{2},\;III=\frac{b_{ij}b_{jk}b_{ki}}{3}$$
where $b_{ij}$ can be given by
(25)$$b_{ij}=\frac{R_{ij}}{R_{kk}}-\frac{1}{3}\delta_{ij}$$$R_{ij}$ is the Reynolds stress tensor, $\delta_{ij}$ is the Kronecker delta tensor, and $R_{kk}$ is twice the turbulent kinetic energy. These invariants can represent the anisotropic “componentiality” of turbulence, which corresponds physically to the relative significance of the three components of fluctuating velocity in a fluid element [22]. The turbulence Lumley triangle map [54,55] is then introduced to characterize the anisotropic behavior within turbulence using only two variables denoted by
(26)$$\eta^{2}=-\frac{II}{2},\;\xi^{3}=\frac{III}{2}$$These variables are plotted in Figure 10a,b for both cases at $x=1.5R_{0}$. The scatter plots are colored by ${\dot{S}}_{\mathrm{heat}}$ to demonstrate the effect of combustion on the anisotropy of turbulence.

In the map, the 1D and 2D vertices denote disk and line shapes of the stress tensor, and the lowest vertex represents spherical shape η=ξ=0 characterizing isotropy. The axisymmetric state is that the variables $\left(\eta ,\xi \right)$ are located at both left and right margins and η=ξ>0 tends to be a “cigar shaped” turbulence state. In the case of NP, it is confirmed that the turbulence without combustion interaction behaves in an axisymmetric manner and move towards the “pancake shaped” state (ξ=−η<0). However, as the magnitude of entropy increases, the couples of values $\left(\eta ,\xi \right)$ lie in the vicinity of the “cigar shaped” turbulence state. In the case of PR, the turbulence in low entropy generation regions is often referred to as the “cigar shaped” turbulence state and is closest to a 2D state, while the evolution towards high entropy regions proceeds along the axisymmetric turbulence state.

### 3.3. Interaction between Reynolds Stress and Combustion Investigated by Entropy

The last section shows that the entropy provides a useful indicator to quantify the complex physics of interaction between vorticity and combustion in the swirling flames. To the best of our knowledge, the entropy generation has not been used for quantifying the interaction between turbulence and combustion before this work. In the following, the Favre-averaged Reynolds stress components of representative $R_{11}$, $R_{22}$ and $R_{12}$, conditionally averaged on the entropy, ${\tilde{\dot{S}}}_{\mathrm{heat}}$, and the normalized radial distance, $\left(r-r_{0}\right)/R_{0}$, are presented in Figure 11a–f. Other components behave similarly and will not be discussed. It is obvious that the Reynolds stress components are positive except shear component $R_{12}$ in the high entropy regions. For low entropy regions, $R_{22}$ is the largest component, but then it starts decaying from its maximum value in high entropy regions. Here and below, we use the term “entropy” instead of the term “entropy generation” for brevity.

In the case of NP (Figure 11a–c), it is observed that $R_{11}$ has a similar trend with increasing ${\tilde{\dot{S}}}_{\mathrm{heat}}$ in the IRZ, MSZ, OSL, and ORZ zones. This trend is observed in both high entropy and low entropy regions. $R_{11}$ reaches its peak in the ISL, which has the highest entropy. This indicates that increasing ${\tilde{\dot{S}}}_{\mathrm{heat}}$ promotes the increase of $R_{11}$. However, $R_{22}$ does not show the same trend in the MSZ, where $R_{22}$ increases because of the decreasing entropy and reaches its maximum magnitude in the OSL. The minimum $R_{12}$ locations correlate with the peak entropy locations in the ISL, while the local $R_{12}$ in the IRZ and MSZ becomes stronger, resulting from the decreasing magnitude of the entropy, peaking in the OSL. Although the Reynolds stress is determined by flow structures, the responses of the components to entropy suggest that combustion plays a significant role in the evolution of the components in swirling flow.

In the case of PR (Figure 11d–f), there are only slight differences in entropy magnitude among the ISL, IRZ, and MSZ regions. This is because entropy generation primarily occurs in the radial direction before the upstream streamwise location $x/R_{0}=1.0$ (see Figure 6b), and the trends of entropy generation profiles along the radial distributions remain almost unchanged after the streamwise locations $x/R_{0}=1.5$. Therefore, the variations of the individual components of $R_{11}$, $R_{22}$, and $R_{12}$ in these intense entropy regions are primarily determined by local flow structures. Note that the components decay in the OSL and ORZ regions as the magnitude of ${\tilde{\dot{S}}}_{\mathrm{heat}}$ decreases. The low shear component $R_{12}$ is generated in strong entropy regions, where $R_{11}$ and $R_{22}$ mainly occur. The increase of $R_{12}$ is observed in the MSZ along the radial distance and reaches the maximum inside the OSL as a result of the strong radial fluctuation velocity and by locally slightly increasing entropy. Following these observations, the Reynolds stress components do not increase completely monotonically with the increasing entropy because swirling structures that determine the Reynolds stress are strongly anisotropic and combustion indicated by the entropy acts to make the turbulence is isotropic in some regions, while there are regions where combustion reinforces anisotropy. The mechanism will be investigated through the analysis of the Reynolds stress budgets.

The Favre-averaged Reynolds stress transport equation is obtained from the momentum Equation (Equation 2),
(27)$$\begin{array}{c} \frac{\partial \bar{\rho }\tilde{{u^{\prime \prime }}_{i}u_{j}^{{}^{\prime \prime }}}}{\partial t}=\underset{T_{1}}{\underset{︸}{-\frac{\partial \bar{\rho }\tilde{u_{k}}\tilde{{u^{\prime \prime }}_{i}{u^{\prime \prime }}_{j}}}{\partial x_{k}}}}\underset{T_{2}}{\underset{︸}{-\frac{\partial \bar{\rho }\tilde{{u^{\prime \prime }}_{k}{u^{\prime \prime }}_{i}{u^{\prime \prime }}_{j}}}{\partial x_{k}}}}\underset{T_{3}}{\underset{︸}{-(\bar{{u_{i}}^{\prime \prime }\frac{\partial p}{\partial x_{j}}}+\bar{{u^{\prime \prime }}_{j}\frac{\partial p}{\partial x_{i}}})}}\\ \underset{T_{4}}{\underset{︸}{-(\bar{\rho }\tilde{{u^{\prime \prime }}_{i}{u^{\prime \prime }}_{k}}\frac{\partial \tilde{u_{j}}}{\partial x_{k}}+\bar{\rho }\tilde{{u^{\prime \prime }}_{j}{u^{\prime \prime }}_{k}}\frac{\partial \tilde{u_{i}}}{\partial x_{k}})}}+\underset{T_{5}}{\underset{︸}{\left(\bar{{u^{\prime \prime }}_{i}\frac{\partial \tau_{jk}}{\partial x_{k}}}+\bar{{u^{\prime \prime }}_{j}\frac{\partial \tau_{ik}}{\partial x_{k}}}\right)}} \end{array}$$
the terms on the RHS represent the different effects on Reynolds stress, corresponding to convective transport $\left(T_{1}\right)$, turbulent transport $\left(T_{2}\right)$, velocity–pressure gradient correlation $\left(T_{3}\right)$, shear production $\left(T_{4}\right)$, and viscous $\left(T_{5}\right)$ terms. It is obviously concluded that in the above discussions, the entropy generation of thermal transport and flow structures together determine the magnitude of the Reynolds stress. For deeply understanding combustion interacting with the Reynolds stress, Figure 12 takes a slice of the radial positions with the highest entropy regions, in which combustion has the strongest interaction with flows. Figure 12a–f shows the budgets of Reynolds stress components, $R_{11}$, $R_{22}$, and $R_{12}$, conditionally averaged on ${\tilde{\dot{S}}}_{\mathrm{heat}}$ in both cases.

Along ${\tilde{\dot{S}}}_{\mathrm{heat}}$, the balance between the velocity–pressure gradient correlation term $\left(T_{3}\right)$ and the shear production term $\left(T_{4}\right)$ primarily determines the development of Reynolds stress, while the viscous term $\left(T_{5}\right)$ is small and insensitive to ${\tilde{\dot{S}}}_{\mathrm{heat}}$. In the case of NP, the convective transport term $\left(T_{1}\right)$ and the shear production term $\left(T_{4}\right)$ act to reinforce positive $R_{11}$ that increases with entropy (Figure 12a). It is noted that the turbulent transport $\left(T_{2}\right)$ varies significantly from positive to negative values along ${\tilde{\dot{S}}}_{\mathrm{heat}}$, indicating that its role is dominated by the combustion effects. Unlike $R_{11}$, the response of the shear production term $\left(T_{4}\right)$ in the budget of $R_{22}$ (Figure 12b) makes the $R_{22}$ lower, which is balanced by the increasing velocity–pressure gradient correlation term $\left(T_{3}\right)$. Therefore, the increasing shear production $\left(T_{4}\right)$ acts to reinforce anisotropy because this plays the opposite role in $R_{11}$ and $R_{22}$. The positive velocity–pressure gradient correlation term $\left(T_{3}\right)$ of $R_{12}$ (Figure 12c) tends to dissipate negative $R_{12}$, and $R_{12}$ remains negative due to the influence of the larger shear production term $\left(T_{4}\right)$.

In the case of PR, the terms in the Reynolds stress budgets are significantly modulated in the highest entropy regions near the OSL, but the magnitudes of all terms are still small due to the influence of turbulence structures. Therefore, it is necessary to analyze the interaction between combustion and the primary occurrence regions of these terms in the MSZ (see Figure 11d–f), where the entropy values are only slightly lower than the peaks near the OSL. The slice analysis of the MSL is shown in Figure 12d–f, and it is observed that the magnitudes of the velocity–pressure gradient term $\left(T_{3}\right)$ and shear production term $\left(T_{4}\right)$ all increase along ${\tilde{\dot{S}}}_{\mathrm{heat}}$. The shear production term $\left(T_{4}\right)$ having the same sign as $R_{11}$ (Figure 12d) and $R_{12}$ (Figure 12f) is performed to reinforce anisotropy, while this contributes to increasingly weaken $R_{22}$ (Figure 12e).

Converse to the shear production term $\left(T_{4}\right)$, the velocity–pressure gradient correlation term $\left(T_{3}\right)$, which plays a fundamentally opposite role in the development of the Reynolds stress components, represents the effect of pressure and is the main term influenced by the entropy in the Reynolds stress budgets. In both cases, the velocity–pressure gradient term $\left(T_{3}\right)$ is high and sensitive to ${\tilde{\dot{S}}}_{\mathrm{heat}}$. To investigate the performances of the normal Reynolds stress, the velocity–pressure gradient term $\left(T_{3}\right)$ can be decomposed into mean pressure terms $\left(T_{3,ii}^{\mathrm{resol}}\right)$ and fluctuating pressure terms $\left(T_{3,ii}^{\mathrm{unresol}}\right)$,
(28)$$\begin{array}{c} -\left(\bar{{u_{i}}^{\prime \prime }\frac{\partial p}{\partial x_{j}}}+\bar{{u^{\prime \prime }}_{j}\frac{\partial p}{\partial x_{i}}}\right)=\underset{T_{3,ii}^{\mathrm{resol}}}{\underset{︸}{-(\bar{{u_{i}}^{\prime \prime }}\frac{\partial \bar{p}}{\partial x_{j}}+\bar{{u_{j}}^{\prime \prime }}\frac{\partial \bar{p}}{\partial x_{i}})}}\underset{T_{3,ii}^{\mathrm{unresol}}}{\underset{︸}{-(\bar{{u_{i}}^{\prime \prime }\frac{\partial p^{\prime }}{\partial x_{j}}}+\bar{{u_{j}}^{\prime \prime }\frac{\partial p^{\prime }}{\partial x_{i}}})}} \end{array}$$Figure 13 shows that the mean pressure terms $\left(T_{3,ii}^{\mathrm{resol}}\right)$ are in a much smaller magnitude than the fluctuating pressure terms $\left(T_{3,ii}^{\mathrm{unresol}}\right)$. The former has negligible contribution and response along ${\tilde{\dot{S}}}_{\mathrm{heat}}$, while the latter is sensitive to ${\tilde{\dot{S}}}_{\mathrm{heat}}$ and even exceeds their sum. This illustrates that the correlation between the velocity fluctuation $\left(u_{1}^{\prime \prime }\right)$ and fluctuating pressure gradient $\left(\frac{\partial p^{\prime }}{\partial x_{1}}\right)$ determines the magnitude of $T_{3}$. In the streamwise direction, the velocity–pressure gradient correlation term $\left(T_{3,11}\right)$ has dominant and negative values due to the large fluctuating pressure gradient $\left(\frac{\partial p^{\prime }}{\partial x_{1}}<0\right)$ with the same sign. This decreasing combustion effect (shown by the entropy in Figure 3) along the streamwise direction generates a negative fluctuating pressure gradient. Therefore, the fluctuating pressure gradient acts to weaken the influence of the shear production term $\left(T_{4}\right)$ in the positive $R_{11}$ budget. From the $T_{3,\mathrm{ii}}^{unresol}$ of Equation (Equation 28), the $\frac{\partial p^{\prime }}{\partial x_{1}}$ in $\bar{{u_{1}}^{\prime \prime }\frac{\partial p^{\prime }}{\partial x_{1}}}$ is negative, so the velocity fluctuation $\left(u_{1}^{\prime \prime }<0\right)$ is also negative but plays an indirect role in this term. Conversely, the velocity–pressure gradient correlation term $\left(T_{3,22}\right)$ acts as the source in the positive $R_{22}$ budget, which is primarily due to the fluctuating pressure gradient. The increase along ${\tilde{\dot{S}}}_{\mathrm{heat}}$ shows that the flame surfaces tend to generate negative fluctuating pressure gradients $\left(\frac{\partial p^{\prime }}{\partial x_{2}}<0\right)$ in the flame-normal direction due to thermal expansion, while the corresponding velocity fluctuation $\left(u_{2}^{\prime \prime }>0\right)$ is positive in sign.

If the fluctuating pressure gradient is generated by velocity fluctuation, there might be ${u_{i}}^{\prime \prime }\frac{\partial p^{\prime }}{\partial x_{i}}>0$. This suggests that the kinetic energy in $u^{\prime \prime }$ is being converted to potential energy stored in the pressure [56]. If the stored potential energy is being converted to kinetic energy, then ${u_{i}}^{\prime \prime }\frac{\partial p^{\prime }}{\partial x_{i}}<0$. The behavior of pressure and kinetic energy are closely correlated. In order to further study the effect of pressure on kinetic energy, the velocity–pressure gradient correlation term $\left(T_{3}\right)$ can be further decomposed into redistribution terms $\left(\psi_{ij}\right)$ and an isotropic term $\left(\phi \right)$,
(29)$$\begin{array}{c} -\left(\bar{{u_{i}}^{\prime \prime }\frac{\partial p}{\partial x_{j}}}+\bar{{u^{\prime \prime }}_{j}\frac{\partial p}{\partial x_{i}}}\right)=\left(\underset{\psi_{ij}}{\underset{︸}{-\bar{{u_{i}}^{\prime \prime }\frac{\partial p}{\partial x_{j}}}-\bar{{u^{\prime \prime }}_{j}\frac{\partial p}{\partial x_{i}}}+\frac{2}{3}\bar{{u^{\prime \prime }}_{k}\frac{\partial p}{\partial x_{k}}}\delta_{ij}}}\right)\underset{\phi }{\underset{︸}{-\frac{2}{3}\bar{{u^{\prime \prime }}_{k}\frac{\partial p}{\partial x_{k}}}\delta_{ij}}} \end{array}$$
the isotropic term $\left(\phi \right)$ is the trace of the velocity–pressure gradient correlation term $\left(T_{3}\right)$, which has direct contribution to the TKE evolution [17]. However, the redistribution terms $\left(\psi_{ij}\right)$ cannot change the energy, and hence they are responsible for intercomponent redistribution [23]. The decomposition of the velocity–pressure correlation term $\left(T_{3}\right)$ in Equation (Equation 29) is shown in Figure 14. It is obvious that the isotropic term $\left(\phi \right)$ progressively approaches zero by increasing entropy. Furthermore, the velocity–pressure gradient correlation term $\left(T_{3}\right)$ in the TKE transport decreases in high entropy regions, which denotes that combustion destroys kinetic energy. For the redistribution terms $\left(\psi_{ij}\right)$, it is noted that the magnitude of each term increases, consistent with the increasing entropy, suggesting that combustion promotes the energy transfer. The streamwise component of the redistribution term $\left(\psi_{11}\right)$ is negative, while other components of the redistribution term $\left(\psi_{22},\psi_{33}\right)$ are positive. It is clear that the velocity–pressure gradient correlation term $\left(T_{3}\right)$ isotropizes turbulence through the redistribution terms by transferring energy from the largest streamwise component $\left(\psi_{11}\right)$ to the other smaller normal components $\left(\psi_{22},\psi_{33}\right)$. Therefore, the trend of weakening anisotropy is also driven by the thermal transport.

The competition between the velocity–pressure gradient correlation term $\left(T_{3}\right)$ and the shear production term $\left(T_{4}\right)$ determines the behavior of the Reynolds stress components in the turbulence field. In both cases, the sign of the shear production term $\left(T_{4}\right)$ is opposite to that of the velocity–pressure gradient correlation term $\left(T_{3}\right)$ [16,17,18], and the influence of thermal expansion through the isotropic term via the shear production term $\left(T_{4}\right)$ needs to be considered and cannot simply be attributed to isotropize turbulence. The isotropic term of the shear production term $\left(T_{4}\right)$ can be represented as the sum of a symmetric part (Ω) and an antisymmetric part (K) [57],
(30)$$\begin{array}{c} -\bar{\rho }\tilde{{u^{\prime \prime }}_{i}{u^{\prime \prime }}_{j}}\frac{\partial \tilde{u_{i}}}{\partial x_{j}}=\underset{\Omega }{\underset{︸}{-\bar{\rho }\tilde{{u^{\prime \prime }}_{i}{u^{\prime \prime }}_{j}}\cdot {\tilde{S}}_{ij}}}\underset{K}{\underset{︸}{-\bar{\rho }\tilde{{u^{\prime \prime }}_{i}{u^{\prime \prime }}_{j}}\epsilon_{ijk}{\tilde{w}}_{k}}} \end{array}$$
where $-\bar{\rho }\tilde{{u^{\prime \prime }}_{i}{u^{\prime \prime }}_{j}}\cdot \tilde{S_{ij}}$ denotes the symmetric part (Ω), which is the major contribution to the isotropic term of the shear production term $\left(T_{4}\right)$. Furthermore, the magnitude of this part is significantly influenced by the alignment of eigenvector pairs. The eigenvectors corresponding to mean strain rate ${\tilde{S}}_{ij}$ are termed the most extensive $s_{1}$, intermediate $s_{2}$, and the most contracting $s_{3}$, respectively. Their eigenvalues $s_{i}$ are ordered $s_{1}>s_{2}>s_{3}$. Similarly, $r_{1}$, $r_{2}$, and $r_{3}$ are the eigenvectors of the Reynolds stress tensor related to the eigenvalues $r_{1}>r_{2}>r_{3}$, which are termed the most extensive, intermediate, and the most contracting principal directions, respectively. The Boussinesq criterion has been demonstrated to be valid in both cases that the most extensive strain-rate eigenvector $s_{1}$ preferentially aligns with the most contracting Reynolds stress eigenvector $r_{3}$, as well as the intermediate eigenvector $r_{2}$ with $s_{2}$, and the most contracting mean strain-rate eigenvector $s_{3}$ with the most extensive Reynolds stress eigenvector $r_{1}$ [22]. However, this alignment relationship interacting with the intense combustion cannot be ignored and is sensitive to the entropy.

The scatter plots of the absolute values of $\left|\cos(r_{2},s_{2})\right|$ and $\left|\cos(r_{3},s_{1})\right|$ colored by the entropy $\left({\tilde{\dot{S}}}_{\mathrm{heat}}\right)$ for the non-premixed flame at $x=1.5R_{0}$ are presented in Figure 15, where the black scatter plots of large $\left|\cos(r_{1},s_{3})\right|>0.8$ are superimposed. It is shown that the high entropy mainly occurs in the regions with $|\cos\left(r_{3},s_{1}\right)|\approx |\cos\left(r_{2},s_{2}\right)|$ denoted by the diagonal area in Figure 15, where the preferential alignment of $r_{1}$ and $s_{3}$ are in good agreement ($\left|\cos(r_{1},s_{3})\right|>0.8$). Therefore, the combustion can apparently contribute to the preferential alignment of eigenvector pairs, i.e., $r_{1}$ and $s_{3}$, $r_{2}$ and $s_{2}$, and $r_{3}$ and $s_{1}$, which directly move the symmetric part (Ω) in Equation (Equation 30) towards higher values. Furthermore, the symmetric part (Ω) determined by the correlation between the Reynolds stress tensor $\tilde{{u^{\prime \prime }}_{i}{u^{\prime \prime }}_{j}}$ and the mean strain-rate tensor $\tilde{S_{ij}}$ peaks in the regions, where the eigenvector pairs are in a full alignment denoted as the upper right area with high entropy in Figure 15. As a result, the shear production term $\left(T_{4}\right)$ is responsible for TKE evolution increases with the entropy.

## 4. Conclusions

The analysis first discusses behaviors of the entropy generation in the non-premixed and premixed flames, showing that the local entropy generation distributes in shear layers and decreases along the streamwise direction. The thermal transport process and its driving force and temperature gradient have a dominant contribution to the entropy generation, which peaks in the ISL for the non-premixed flame while in the MSZ for the premixed flame. The flame distributes more broadly in the premixed flame due to a great thermal transport in the radial direction, while a stronger but narrower flame distribution is caused by the intense chemical reaction process in the non-premixed flame.

The interaction between the enstrophy dynamics and combustion in both flames is quantified by the entropy due to the thermal transport and its driving force temperature gradient. The balance between the vortex stretching and the dissipation determines the evolution of enstrophy, and these individual terms are found to have a similar trend with increasing entropy. A joint PDF analysis of the vortex stretching with entropy demonstrates that the enstrophy is produced in high entropy regions while it is accompanied by an increasing dissipation. In the non-premixed flame, the turbulence anisotropy in the Lumley triangle leaves the “pancake shaped” state and moves towards the “cigar shaped” state with increasing entropy. In the premixed flame, the turbulence state with low entropy is referred to as the “cigar shaped” state, while the evolution towards high entropy regions proceeds along the axisymmetric state.

The Reynolds stress is finally examined, and substantial increases of its normal components $\left(R_{ii}\right)$ with the entropy are observed. The shear components are determined by flow structures, and they peak in the OSL. The budget analysis shows that the velocity–pressure gradient and shear production terms dominate the Reynolds stress and are sensitive to the entropy. The velocity–pressure gradient correlation term acts to reinforce $R_{22}$ and increases with the entropy, while it dissipates positive $R_{11}$. This is determined by the correlations of the fluctuating pressure gradient and the entropy. Further analysis finds that the velocity–pressure gradient correlation term isotropizes turbulence by transferring energy from the largest streamwise component ($R_{11}$) to the other two normal components of Reynolds stress. Evidence shows that the combustion increases the preferential alignment of the most contracting mean strain-rate eigenvector $s_{3}$ with the most extensive Reynolds stress eigenvector $r_{1}$, resulting in a greater shear production term in the TKE transport.

In summary, results in this present work provide an insightful view into the physics between the turbulence and combustion of the high pressure swirling non-premixed and premixed flames. The DNS results suggest that the entropy directly measuring the thermodynamic process in the combustion system can serve as a useful piece of information for interaction between combustion and turbulence.

## Figures and Tables

**Figure 1 entropy-25-01151-f001:**
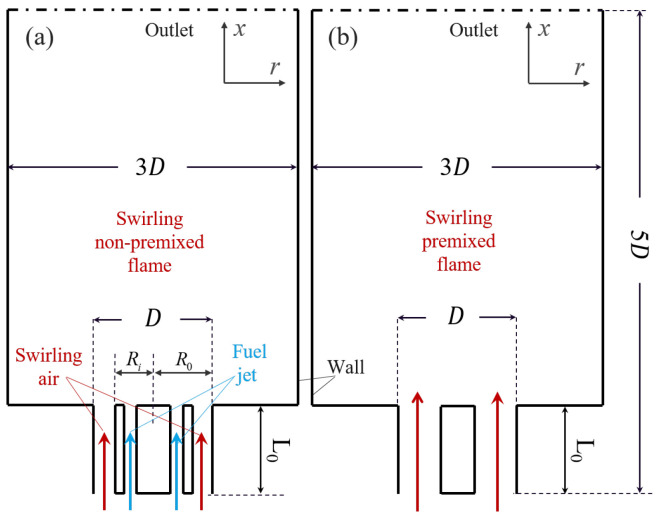
DNS burners structure of the non-premixed (**a**) and premixed (**b**) cases.

**Figure 2 entropy-25-01151-f002:**
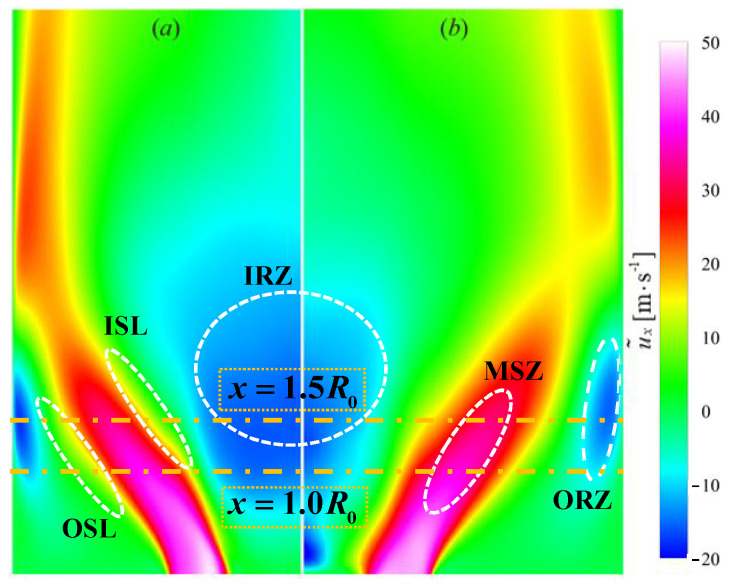
Distribution of the mean axial velocity for the non-premixed (**a**) and premixed (**b**) cases.

**Figure 3 entropy-25-01151-f003:**
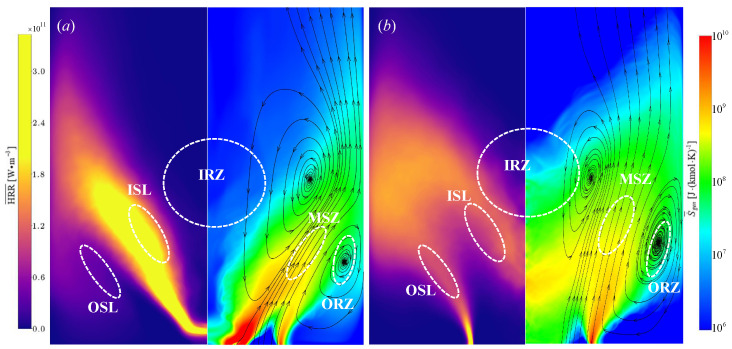
Contours of Reynolds-averaged heat release (**left**) and entropy generation (**right**) distributions for the non-premixed (**a**) and premixed (**b**) flames. The black line indicates streamlines obtained from time-averaged velocity.

**Figure 4 entropy-25-01151-f004:**
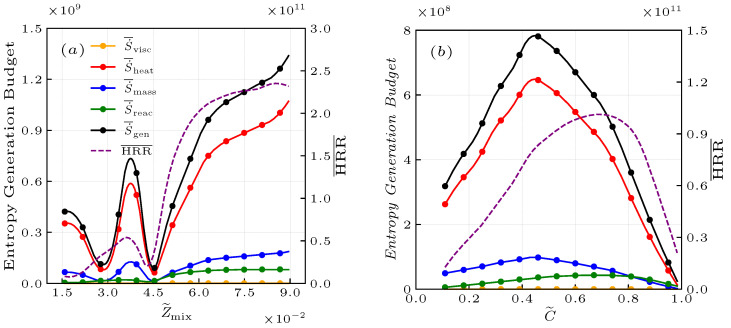
Budget of the entropy generation and heat release conditionally averaged on the ${\tilde{Z}}_{mix}$ for the non-premixed (**a**) and $\tilde{C}$ for premixed (**b**) flames at $x/R_{0}=1.0$.

**Figure 5 entropy-25-01151-f005:**
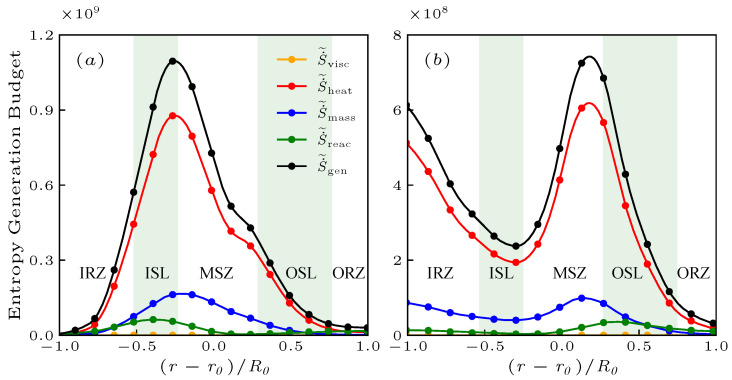
The radial distribution of the entropy generation budget for the non-premixed (**a**) and $\tilde{C}$ premixed (**b**) flames at $x/R_{0}=1.0$.

**Figure 6 entropy-25-01151-f006:**
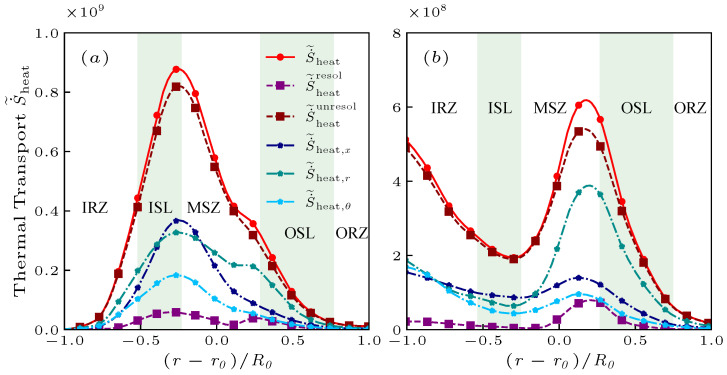
The radial distribution of ${\tilde{\dot{S}}}_{\mathrm{heat}}$ and its components in the streamwise direction (${\tilde{\dot{S}}}_{\mathrm{heat},x}$), radial direction (${\tilde{\dot{S}}}_{\mathrm{heat},r}$), and circumferential direction (${\tilde{\dot{S}}}_{\mathrm{heat},\theta }$) of the non-premixed (**a**) and premixed (**b**) flames at $x/R_{0}=1.0$. Additionally, the resolved component (${\tilde{\dot{S}}}_{\mathrm{heat}}^{\mathrm{resol}}$) and unresolved component (${\tilde{\dot{S}}}_{\mathrm{heat}}^{\mathrm{unresol}}$) of Equations (Equation 21) and (Equation 22) are presented.

**Figure 7 entropy-25-01151-f007:**
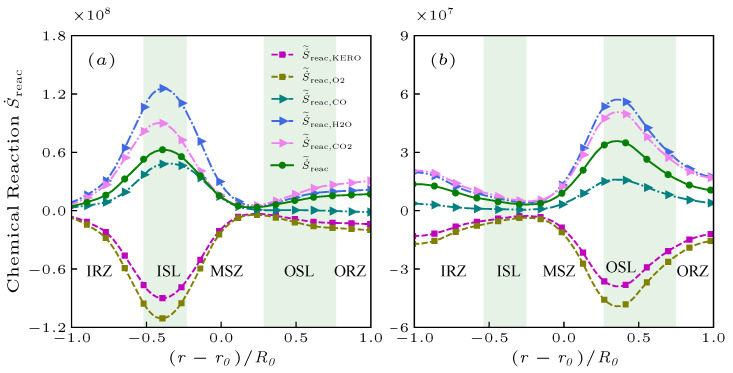
The radial distribution of the ${\tilde{\dot{S}}}_{\mathrm{reac}}$ and its components for the non-premixed (**a**) and premixed (**b**) flames at $x/R_{0}=1.0$.

**Figure 8 entropy-25-01151-f008:**
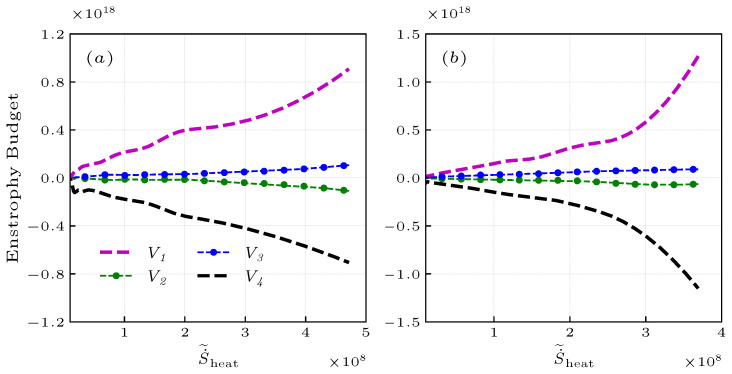
Budget of the enstrophy transport conditionally averaged on ${\tilde{\dot{S}}}_{\mathrm{heat}}$ for the non-premixed (**a**) and premixed (**b**) flames at $x/R_{0}=1.5$.

**Figure 9 entropy-25-01151-f009:**
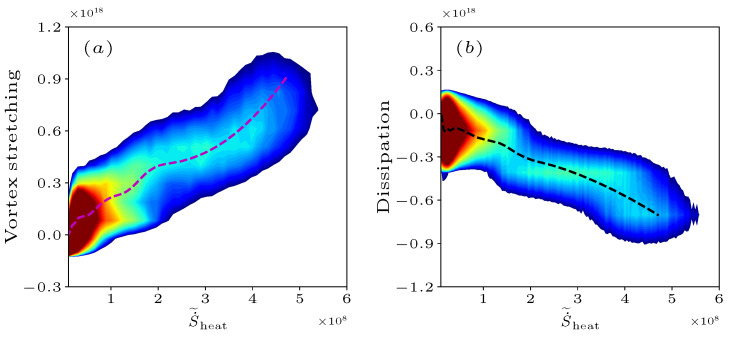
The joint PDF of (**a**) the vortex stretching $\left(V_{1}\right)$ and (**b**) the dissipation $\left(V_{4}\right)$ in Equation (Equation 23) along with ${\tilde{\dot{S}}}_{\mathrm{heat}}$ for the non-premixed flame at $x=1.5R_{0}$, with the conditional mean indicated by the dotted line.

**Figure 10 entropy-25-01151-f010:**
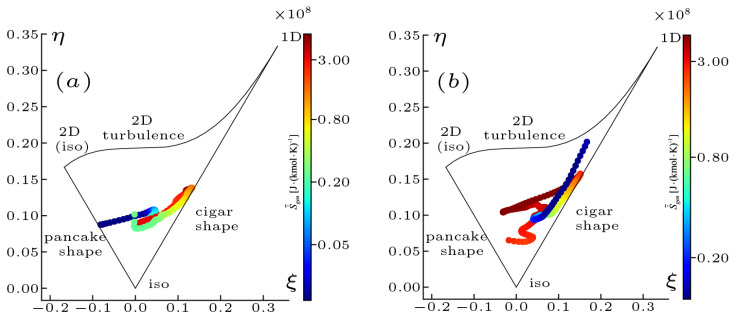
Maps of the Reynolds stress anisotropy invariants η and ξ plotted in the Lumley triangle [54] for the non-premixed (**a**) and premixed (**b**) flames at $x=1.5R_{0}$. The scatter plots colored by ${\dot{S}}_{\mathrm{heat}}$ are also shown.

**Figure 11 entropy-25-01151-f011:**
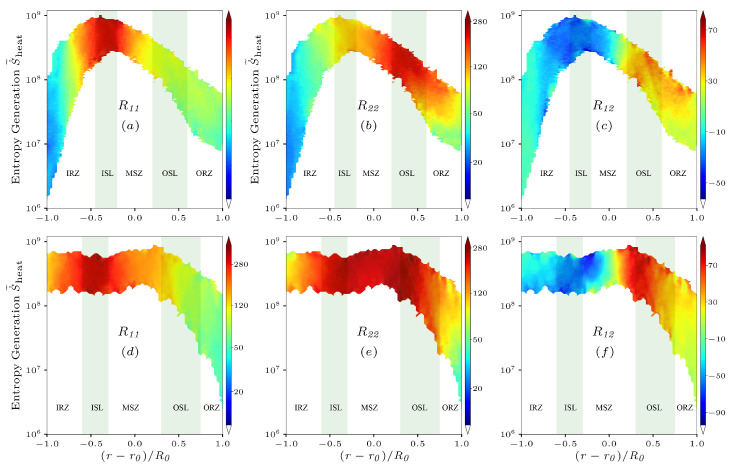
Reynolds stress components $R_{11}$, $R_{22}$, and $R_{12}$ conditionally averaged on the entropy, ${\tilde{\dot{S}}}_{\mathrm{heat}}$, and the normalized radial distance, $\left(r-r_{0}\right)/R_{0}$, for the non-premixed (**a**–**c**) and premixed (**d**–**f**) flames at $x=1.5R_{0}$.

**Figure 12 entropy-25-01151-f012:**
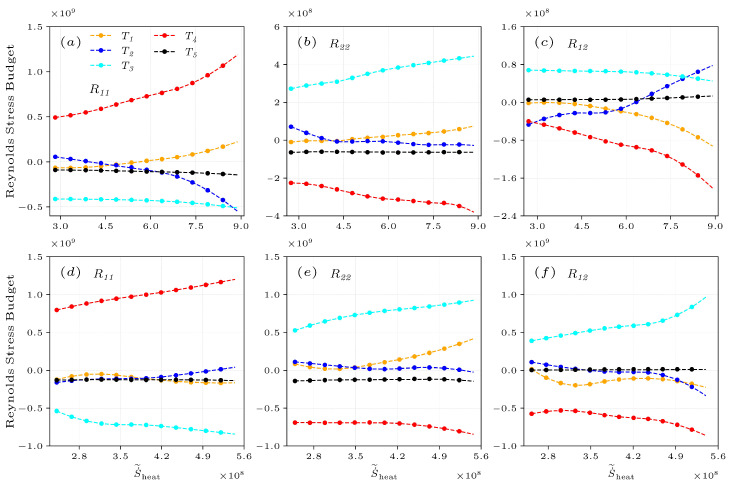
Reynolds stress budgets of $R_{11}$, $R_{22}$, and $R_{12}$ conditionally averaged on ${\tilde{\dot{S}}}_{\mathrm{heat}}$ for the non-premixed (**a**–**c**) and premixed (**d**–**f**) flames at $x=1.5R_{0}$.

**Figure 13 entropy-25-01151-f013:**
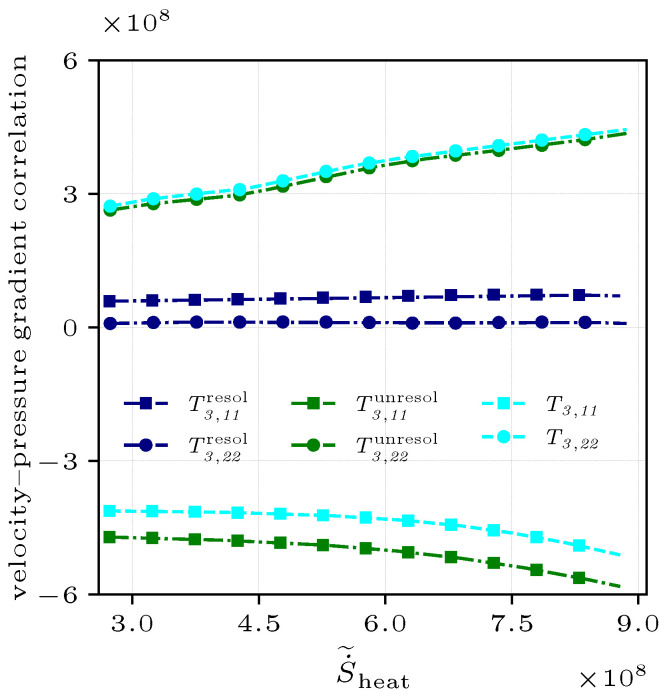
Decomposition of the velocity–pressure gradient correlation term $\left(T_{3}\right)$ into the mean pressure $\left(T_{3,ii}^{\mathrm{resol}}\right)$ and fluctuating pressure $\left(T_{3,ii}^{\mathrm{unresol}}\right)$ terms for the non-premixed flame at $x=1.5R_{0}$.

**Figure 14 entropy-25-01151-f014:**
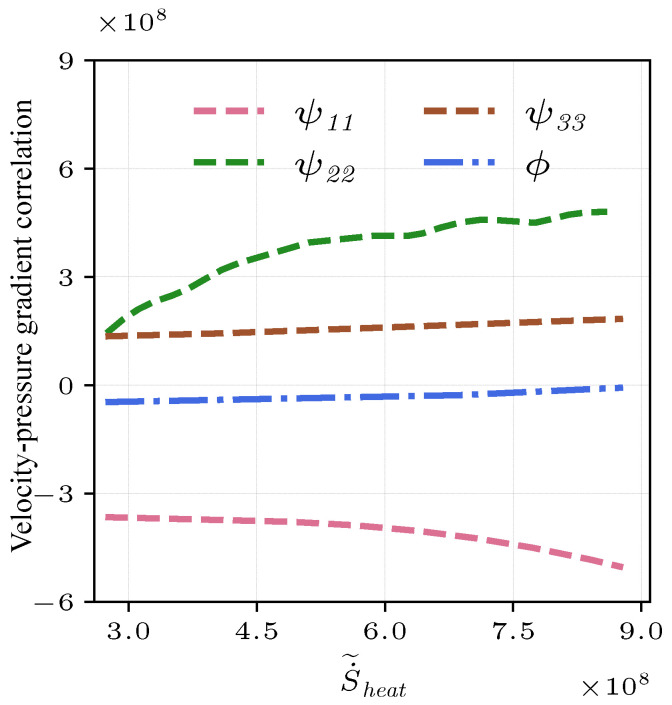
Decomposition of the velocity–pressure gradient correlation term $\left(T_{3}\right)$ into the redistribution $\left(\psi_{ij}\right)$ and the isotropic $\left(\phi \right)$ terms for the non-premixed flame at $x=1.5R_{0}$.

**Figure 15 entropy-25-01151-f015:**
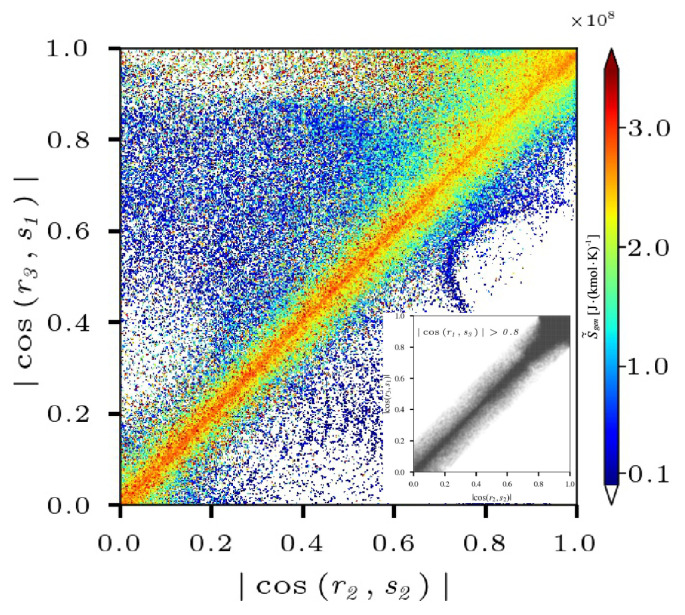
The scatter plots of $\left|\cos(r_{2},s_{2})\right|$ and $\left|\cos(r_{3},s_{1})\right|$ colored by ${\tilde{\dot{S}}}_{\mathrm{heat}}$ for the non-premixed flame at $x=1.5R_{0}$, where the black scatter plots of large $\left|\cos(r_{1},s_{3})\right|>0.8$ are superimposed.

## Data Availability

The data that support the findings of this study are available from the corresponding author upon reasonable request.

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
