# Peer review of "Entropy: An Inspiring Tool for Characterizing Turbulence–Combustion Interaction in Swirling Flames via Direct Numerical Simulations of Non-Premixed and Premixed Flames"

_entropy, 2023, doi:10.3390/e25081151_

Round 1

Author Response

We are grateful for the reviewer's constructive comments which helped us to improve the manuscript.  The response to the comments is provided in the document below.

Reviewer 2 Report

Good work !

Recommend for the publication. 

Author Response

We greatly appreciate your acknowledgment of our work.  Best regards!

Reviewer 3 Report

The Authors propose a research to investigate the role of entropy generation in turbulent combustion processes. In particular, they address the link  etween the generation of entropy and turbulence. The study indicates that
a dominant role in entropy generation is played by thermal transport (driven by temperature gradients). In addition, the analysis highlights the impact of entropy on turbulence characteristics. In this regard, the work deeply investigates from a computational point of view two test cases structured on similar geometries. Hence, both non-premixed and premixed combustion are studied. The research is interesting and should be considered
for publication upon some major adjustments:

1) English should be  revised to favor the readability and the understanding. Few examples are: lines 114-116, 144-145, 171, 178-189, 234-235, 409-411.

2) In Section 2, the Authors describe the test cases before presenting the governing equations. Presenting the governing equations first, and having a subsequent section for the presentation of the test cases would ease the readability.

3) Eq. 11 can be written occupying one single line. Same for Eq. 12.

4) In all figures having a vertical colored bar, please add on the side the symbol related to the visualized field.

5) Use Fig. 1 or Fig. 2 to indicate where x/R0 = 1.0 and 1.5 are located. Also, revise the other figure captions
to point to the chart where x/R0 = 1.0 and 1.5 are indicated.

6) (OPTIONAL) Fig. 4b is discussed together with $\tilde{C}$ before Fig. 4a. To preserve consistency in the text, please
invert the discussion of Fig. 4a and 4b.

7) Define $\tilde{C}$ and $\tilde{Z}_{mix}$ using two equations, instead of presenting them within the text.

8) In Fig. 5, 6, and 7 please consider to add a legend on the side (right, external) of the two subfloats, so  you can preserve the same wording (IRZ, ISL, ..., ORZ) in both subfloat a and b. Also, in the caption add a reference to Fig. 2, where IRZ, ISL, ..., ORZ are reported graphically.

9)  In lines 280-282 the Authors state that the generation of entropy for the premixed case mainly occurs in IR and OSI regions. Nonetheless, Fig. 5 and 6 show peaks in regions IRZ and MSZ. This is not very clear to me.

10) Present the resolved and unresolved terms of the entropy $\dot{S}_{heat}$ (lines 283-286) using two equations,
instead of presenting them within the text. In this way, the equations can be referenced in the figures' caption to favor comprehension. 

11) In the caption of Fig. 9 please add a reference to Eq. 16 so that one can immediately point out V1 and V4.

12) In lines 380-387 please consider defining the tensor and the invariants using equations, instead of presenting them within the text.

13) In the maps of Fig. 10, the understanding is made complex by the coloring of the points using $\dot{S}_{heat}$.
Please add a comment in the text (lines 388-397) to better clarify this aspect.

14) In lines 409-411 consider changing ``increases'' with ``has a similar trend''.

15) In Fig. 11, please add the wording IRZ, ISL, ..., ORZ in all the sub-figures.

16) In line 427 consider changing ``near'' with ``inside''.

17) In the conclusive section 4, please revise lines 554 (``OSL'') and 561 (``increase'') based on the suggestions proposed in points 9 and 14.

18) In the References section, add the doi of the cited literature works.

The English is fine, but the way the paper is organized should be improved to read it more easily.

Author Response

(The authors gave the same response as above.)

Author Response

(The authors gave the same response as above.)

Round 2

Reviewer 3 Report

Authors did the required modifications

Author Response

We would like to express our sincere gratitude to the reviewer for valuable feedback and insightful suggestions. The expertise and constructive comments have played a crucial role in improving the quality and clarity of this paper.

Reviewer 4 Report

Many thanks for satisfactory answers to my 30 remarks. I have only two minor recommendations that can be taken into account without my further review. These recommendations are given in the attached file.

Author Response

(The authors gave the same response as above.)
